# Longitudinal Association of Telomere Dynamics with Obesity and Metabolic Disorders in Young Children

**DOI:** 10.3390/nu14235191

**Published:** 2022-12-06

**Authors:** Simon Toupance, Sofia I. Karampatsou, Carlos Labat, Sofia-Maria Genitsaridi, Athanasia Tragomalou, Penio Kassari, George Soulis, Allyson Hollander, Evangelia Charmandari, Athanase Benetos

**Affiliations:** 1Université de Lorraine, Inserm, DCAC, F-54000 Nancy, France; 2Division of Endocrinology, Metabolism and Diabetes, First Department of Pediatrics, National and Kapodistrian University of Athens Medical School, ‘Aghia Sophia’ Children’s Hospital, 11527 Athens, Greece; 3Outpatient Geriatric Assessment Unit, Henry Dunant Hospital Center, 11526 Athens, Greece; 4Division of Endocrinology and Metabolism, Center for Clinical, Experimental Surgery and Translational Research, Biomedical Research Foundation of the Academy of Athens, 11527 Athens, Greece; 5Université de Lorraine, CHRU-Nancy, Pôle “Maladies du Vieillissement, Gérontologie et Soins Palliatifs”, F-54000 Nancy, France

**Keywords:** telomeres, obesity, metabolic disorders, determinants, children, childhood

## Abstract

In adults, short leukocyte telomere length (LTL) is associated with metabolic disorders, such as obesity and diabetes mellitus type 2. These associations could stem from early life interactions between LTL and metabolic disorders. To test this hypothesis, we explored the associations between LTL and metabolic parameters as well as their evolution over time in children with or without obesity at baseline. Seventy-three (*n* = 73) children attending our Outpatient Clinic for the Prevention and Management of Overweight and Obesity in Childhood and Adolescence, aged 2–10 years (mean ± SD: 7.6 ± 2.0 years), were followed for 2 to 4 years. Anthropometric, clinical, and biological (including LTL by Southern blot) measurements were performed annually. Baseline LTL correlated negatively with BMI (*p* = 0.02), fat percentage (*p* = 0.01), and blood glucose (*p* = 0.0007). These associations persisted after adjustments for age and sex. No associations were found between LTL attrition during the follow-up period and any of the metabolic parameters. In young children, obesity and metabolic disturbances were associated with shorter telomeres but were not associated with more pronounced LTL attrition. These results suggest that short telomeres contribute to the development of obesity and metabolic disorders very early in life, which can have a major impact on health.

## 1. Introduction

Telomere length (TL), one of the primary hallmarks of aging [1], decreases with age and is associated with age-related diseases [2]. In adults, short leukocyte telomere length (LTL) is associated with metabolic disorders, such as obesity, insulin resistance, and diabetes mellitus type 2 [3,4]. The causality of these associations remains unclear since longitudinal studies do not show an influence of these metabolic disorders on LTL attrition during adult life [5,6]. Several epidemiological studies in large adult populations, mostly conducted using a Mendelian randomization approach, have shown a causal role of short telomeres in the risk of developing chronic cardio-metabolic diseases [7,8]. 

Obesity and metabolic disorders in young children represent one of the most important public and individual health problems worldwide [9,10,11]. Shorter telomeres are already present in obese children [12]; however, little is known regarding the early-age interactions of obesity and metabolic disorders with TL dynamics (TL and its evolution over time). The potential effect of obesity on telomere attrition in a critical period of growth and proliferative activity or the causal role of preexisting short telomeres on obesity and metabolic disorders development could explain the associations between these parameters. In the present longitudinal study, we determined the LTL, metabolic parameters, and their respective evolution over a period of 2 to 4 years in children with or without obesity at baseline, aged 2–10 years at inclusion.

## 2. Methods

### 2.1. Patients

Seventy-three (n = 73) children aged 2–10 years (mean age ± SD: 7.6 ± 2.0 years; 25 males, 48 females) attending our Outpatient Clinic for the Prevention and Management of Overweight and Obesity in Childhood and Adolescence, Division of Endocrinology and Metabolism, First Department of Pediatrics, ‘Aghia Sophia’ Children’s Hospital, were studied prospectively for a period of up to 4 years. The inclusion criteria were as follows: (i) aged 10 or less at inclusion; (ii) written informed consent provided by their parents. Patients were recruited in 2014 and 2015 and followed until 2019. Patients with a normal BMI at baseline were self-referred to our outpatient clinic in order to obtain information on lifestyle interventions to prevent obesity. Patients with overweight and simple obesity were either self-referred to our outpatient clinic or referred by their pediatricians and/or general practitioners on account of their obesity. Patients with syndromic or monogenic obesity as well as patients who required psychological or psychiatric input were excluded from the study. Subjects were classified as having obesity, overweight, or normal BMI according to the International Obesity Task Force (IOTF) cut-off points [13].

### 2.2. Clinical Examination

All participants were admitted to the Endocrine Unit early in the morning on the day of the study, and a detailed medical history and clinical examination, including pubertal assessment and standard anthropometric measurements (weight, height, waist circumference, and hip circumference) were obtained by a single trained observer. Body weight was measured in light clothing and without shoes using the same scale for all subjects (Seca GmbH & Co. KG., Hamburg, Germany). Standing height was also measured without shoes using a stadiometer (Holtain Limited, Crymych-Dyfed, UK). Waist and hip circumferences were measured according to the World Health Organization (WHO) STEPS protocol using the same stretch-resistant tape (Seca GmbH & Co. KG., Hamburg, Germany) with the subject in a standing position [14]. More specifically, waist circumference was measured in the horizontal plane midway between the lowest rib and the iliac crest at the end of a normal expiration. Hip circumference was measured in the horizontal plane at the level of the maximum circumference of the hips and buttocks. On both occasions, the tape did not compress the skin and was parallel to the floor. In addition, all subjects underwent bioelectrical impedance analysis (BIA) (TANITA MC-780U Multi Frequency Segmental Body Composition Analyzer, Amsterdam, The Netherlands) [15,16].

A blood sample for baseline hematological, biochemical, and endocrinological investigations was collected from the antecubital vein at 8 a.m. following a 12-h fast. Samples were centrifuged and separated immediately after collection and were stored at −80 °C until assayed.

### 2.3. Assessment and Interventions

At the initial assessments, all subjects were evaluated by a pediatrician and pediatric dietitian for their daily eating habits, and a 24-h recall of their diet based on the United States Department of Agriculture (USDA) method was performed [17]. Subsequently, children and their parents were informed about the complications of obesity and the need for the whole family to adopt a healthier lifestyle. A description of the protocol is detailed in [15,16,18]. 

At each subsequent appointment, the pediatrician re-assessed the anthropometric measurements, and the goals set in previous sessions were discussed in detail with the pediatric dietitian and the personal trainer, as well as the possible difficulties faced by children in achieving their optimal BMI. Detailed hematological, biochemical, and endocrinological investigations were performed at least annually (more frequently if required). A blood sample for DNA analysis was also obtained every year [15,16,18].

### 2.4. Assays and Scores

Standard hematological investigations were determined using the ADVIA 2110i analyzer (Roche Diagnostics GmbH, Mannheim, Germany). The concentration of glucose was determined using the ADVIA 1800 Siemens analyzer (Siemens Healthcare Diagnostics, Tarrytown, NY, USA). Insulin was measured using automated electro-chemiluminescence immunoassays (Analyzer Cobas e411, Roche Diagnostics GmbH). Hemoglobin A1c (HbA1c) was determined using reversed-phase cation exchange high-performance liquid chromatography on an automated glycohemoglobin analyzer HA-8160 (Arkray, Kyoto, Japan). 

All assays were performed at the laboratories of ‘Aghia Sophia’ Children’s Hospital using the same methods and equipment during the 4-year follow-up period.

The glucose score, insulin score, Hb1Ac score, waist/hip score, and fat percentage score were defined as the mean values of these parameters throughout the study.

### 2.5. BMI Score and BMI Profiles 

The BMI score was calculated on the basis of the mean BMI category during follow-up. Categories from 0 to 4 were normal BMI = 0, overweight = 1, obesity = 2, and severe obesity = 3, according to the criteria used for young children [13]. Each participant was categorized at each visit, and the BMI score was defined as the mean BMI category throughout the study.

Three BMI profiles were created according to the BMI category at visit 1 of the study and its evolution throughout the study. Profile 1 included all individuals with BMI category 0 during visit 1 (n = 12), profile 2 included individuals with BMI categories > 0 at visit 1 and a decrease in the BMI category over the follow-up period (n = 48), and profile 3 included individuals with BMI category > 0 at visit 1 and an increase in the BMI category throughout the follow-up (n = 13) (Figure 1).

### 2.6. LTL Measurements by Southern Blot

Leukocyte DNA was extracted from frozen buffy coats with a salting-out protocol (Puregene kit, Qiagen, Hilden, Germany) and resuspended in TE 1X buffer. DNA quantity was assessed by spectrophotometry (Nanodrop, Thermo Fisher Scientific, Waltham, MA, USA). DNA samples passed integrity testing using a 1% (wt/vol) agarose gel prior to performing the TL measurement by Southern blot analysis of the terminal restriction fragments, as described previously [19]. Briefly, DNA samples were treated overnight with restriction enzymes HinfI and RsaI (Roche Diagnostics GmbH, Mannheim, Germany). Digested DNA samples and DNA ladder were resolved on 0.5% (wt/vol) agarose gels for 23 h. After depurination, denaturation, and neutralization, DNA was transferred onto a positively charged nylon membrane (Roche Diagnostics GmbH) using a vacuum blotter (Biorad, Hercules, CA, USA). Membranes were hybridized at 42 °C with a digoxigenin-labeled telomeric probe. The probe was subsequently detected by the DIG luminescent detection procedure (Roche Diagnostics GmbH) and exposed on a charge-coupled device camera (Las 4000, Fujifilm Life Sciences, Cambridge, MA, USA). Different samples from the same individual (baseline and follow-up visits) were always run in adjacent lanes on the same membrane. Measurements were performed in duplicate on separate membranes. The inter-assay coefficient of variation for the duplicate measurements was 1.98%.

### 2.7. Statistical Analyses

Continuous variables are presented as means ± SD or means ± SE (in histograms), and discrete variables are presented as percentages. Comparisons of parameters between the 3 BMI profiles were carried out using one-way ANOVA and/or trend ANOVA followed by the Tukey–Kramer post hoc test for comparison between 2 profiles. In the absence of a normal distribution, Kruskal–Wallis one-way ANOVA was used for baseline insulin and baseline waist/hip Bivariate relationships between continuous variables were determined using Pearson’s correlation coefficients. Associations between metabolic phenotypes and telomere dynamics were assessed by multivariate analysis. Only variables with *p* < 0.10 in the univariate analysis were included in the multivariate models. A *p*-value < 0.05 was considered significant. Statistical analyses were performed using the NCSS 9 statistical software package (NCSS, Kaysville, UT, USA).

## 3. Results

### 3.1. Characteristics of the Participants and Classification According to BMI Profiles

The participants’ characteristics are summarized in Table 1. Ages ranged from 2.4 to 10.4 years at inclusion and from 4.7 to 13.9 years at the end of the follow-up period. The mean follow-up duration was 2.8 ± 0.7 years. The mean LTL attrition during this follow-up was 67 ± 8 bp/year. At baseline, differences were observed between the three BMI profiles with regard to gender, waist and hip circumferences, serum insulin concentrations, and fat percentage. There were no differences in standard hematological parameters between the three groups (Data not shown).

### 3.2. LTL Dynamics in the Three BMI Profiles

Age- and sex-adjusted baseline LTL differed in the three profiles (ANOVA, *p* = 0.02). The subsequent two-by-two comparisons showed a longer LTL in profile 1 than in profile 2 (Figure 2a). The same trend was observed for the final follow-up LTL between the three profiles (Pr1: 8.84 ± 0.21, Pr2: 8.26 ± 0.11, Pr3: 8.53 ± 0.20; *p* = 0.053). No difference was observed in LTL attrition between the three profiles (Figure 2b).

### 3.3. Association between LTL Dynamics and Obesity Indicators

Baseline LTL correlated negatively with BMI score (R = −0.26; *p* = 0.02; Figure 3a) and fat percentage score (R = −0.29; *p* = 0.01; Figure 3c). A trend for a negative association was observed between baseline LTL and waist/hip score (*p* = 0.07; Figure 3b). Similar results were observed when LTL values at the end of follow-up were used (*p* = 0.015 for BMI score, *p* = 0.18 for waist/hip score, and *p* = 0.01 for fat percentage score) (data not shown). Multivariate model analyses including age and sex as covariables confirmed negative associations between baseline LTL and BMI score, waist/hip score, and fat percentage score (*p* = 0.02, *p* = 0.005, and *p* = 0.004, respectively; Table 2). No association was found between LTL attrition and BMI score (β = −0.0003; *p* = 0.85), waist/hip score (β = 0.01; *p* = 0.27), or fat percentage score (β = 0.0036; *p* = 0.73) in the univariate analyses or the multivariate models including average age during follow-up and baseline LTL as covariates (data not shown).

### 3.4. Association between LTL Dynamics and Metabolic Disorders

Baseline LTL correlated negatively with blood glucose scores (R = −0.39; *p* = 0.0007; Figure 4a). Trends for negative associations were observed between baseline LTL and insulinemia score (*p* = 0.07; Figure 4b) and Hb1Ac score (*p* = 0.07; Figure 4c). Similar results were observed when LTL values at the end of follow-up were used (*p* = 0.0004 for blood glucose score, *p* = 0.10 for insulinemia score, and *p* = 0.13 for Hb1Ac score). Multivariate model analysis including age and sex as covariables confirmed the negative association between baseline LTL and blood glucose score (*p* = 0.0007; Table 3). LTL attrition was not correlated with blood glucose score (β = −0.012; *p* = 0.19), insulinemia (β = 0.007; *p* = 0.55), or Hb1Ac scores (β = 0.0003; *p* = 0.42) in the univariate analyses or the multivariate models including average age during follow-up and baseline LTL as covariates (Data not shown).

## 4. Discussion

In this longitudinal study on young children, we found that short telomeres were associated with clinical phenotypes of obesity and metabolic disorders. No association was found between these metabolic parameters and LTL attrition over the follow-up period. These results indicate that short telomeres precede the development of childhood obesity and suggest that TL may be involved in the development of obesity and metabolic disorders very early in life, which has a major impact on health.

The links between TL and metabolic disorders, such as obesity [4,20,21,22,23], metabolic syndrome, and diabetes, have been shown in several cohorts of adult subjects [3,24,25]. These findings have led several authors to the conclusion that factors involved in or associated with these metabolic disorders, such as diet, lack of physical activities, chronic inflammation, and high oxidative stress, are responsible for accelerated telomere attrition [26,27]. Thus, according to this concept, short telomeres could theoretically be used as a biomarker to detect the cellular impact of metabolic disorders or even predict the response to obesity treatment [28].

However, clinical longitudinal studies with sequential TL measurements over time have been unable to detect accelerated telomere attrition in adults with metabolic disorders compared with people without such disturbances [5,29,30]. In addition, no relationship has been observed between nutritional factors and telomere attrition [31,32,33,34,35,36].

Such observations raise the hypothesis that short telomeres in adult age may precede the development of metabolic disorders, i.e., they are the cause rather than the consequence of these disorders [6]. In order to further explore this question, we carried out the current longitudinal study with sequential measurements of TL and metabolic parameters in a cohort of very young children with and without obesity at baseline. We found consistent negative associations between LTL and obesity parameters, such as BMI, waist/hip ratio, and fat percentage. The associations were less consistent with metabolic parameters, with a significant negative association for glycemia and only trends for insulinemia and Hb1Ac. These results observed in children with obesity and metabolic disorders indicate that the aforementioned links already exist during the very first years of life. A few other studies have observed links between short telomeres and obesity [12,28,37] as well as increased glucose levels in children [38]. A study also showed the link between early obesity (at age 6 months) and short telomeres measured 3 to 5 years later [39], although telomere length was not measured at the age of 6 months when obesity was observed. The longitudinal design of the present study provides further significant insight in that TL attrition during the follow-up period was not related to the obesity or metabolic status of the children. These results, along with those obtained by the Mendelian randomization approach, indicate that short telomeres precede metabolic disorders and suggest that telomeres may have a causal role in the development of early-life obesity and metabolic disorders in children.

The associations between LTL and obesity were slightly weaker at the end of the follow-up period compared with those observed at baseline. This may be explained either by variable telomere attrition rates in the population or by the development of obesity linked to environmental factors. This may obscure the TL–obesity association in later life, and this is in accordance with the large-scale meta-analysis required to obtain final proof of this association in adults [4]. Interestingly, if true, this notion will heighten the causal role of TL since obesity-enhanced TL attrition would strengthen the TL–obesity association.

In addition, studies in humans in adulthood have already highlighted the role of telomeres as a predictor of response to diet [40], weight gain [41], the development of insulin resistance [6], and diabetes [42,43,44]. In children, TL is predictive of the outcome of lifestyle intervention on glucose levels [45] and obesity [46]. The LTL at age 5 years is also partially predictive of obesity at 9 years [47], and shorter LTL during childhood is associated with higher triglyceride levels in early adolescence [48]. Longer LTL has even been described in adults as a predictor of positive response to a behavioral intervention [49].

The exact mechanisms of the role of TL as a modulator of the effects of diet on metabolic disorders are still unknown, although some studies have proposed certain mechanistic hypotheses. Elks and Scott proposed that short telomeres could induce pancreatic β-cell senescence, leading to impaired glucose tolerance, in turn leading to diabetes [50]. It has also been suggested that short TL leads to adipocyte senescence, senescence-associated secretory phenotype (SASP), changes in adipocyte metabolic profiles, and less adipogenesis [51,52].

Although the current results are in favor of a causal role of TL in the development of metabolic disorders, we cannot eliminate a possible role of inflammatory, nutritional, and metabolic factors even earlier in life on the regulation of telomere length, especially during pregnancy [53]. Lazarides et al. found shorter TL in newborns from mothers with a pro-inflammatory state [54], and Wei et al. reported a relationship between maternal smoking and short TL in children [55]. Studies have, furthermore, shown an inverse relationship between the BMI of the mother before pregnancy and the TL of the child [56,57], as well as between gestational diabetes and shorter TL in offspring [58,59,60], although these studies did not control for the TL of the mother. McAninch et al. also found an association between metabolic syndrome in pregnancy and reduced TL in children at age 10 [61].

Certain limitations of our study should be acknowledged. The sample size and the duration of the follow-up of this study are limited, and thus we cannot eliminate a lack of statistical power to detect differences in attrition rates among the different obesity profiles. Moreover, the potential faster attrition in the youngest children compared with the oldest ones in our study (age range of 8 years) could obscure the hypothetical impact of obesity on attrition, and adjusting for this phenomenon is hard given the limited sample size as well as the potentially non-linear effects.

In conclusion, the results of the present longitudinal study in young children indicate that short telomeres may contribute to the development of obesity and metabolic disorders very early in life, thus having a major impact on health. If these results are confirmed in larger cohorts, telomere length could be considered a risk factor for early-life obesity and could be used to propose personalized interventions for children at risk.

## Figures and Tables

**Figure 1 nutrients-14-05191-f001:**
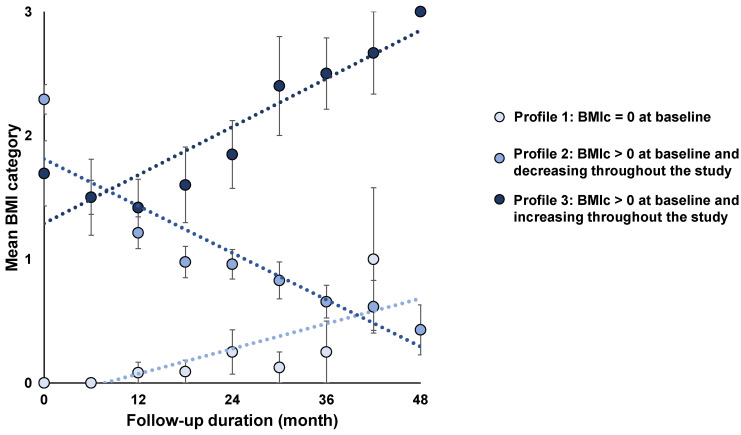
BMI profiles according to BMI categories at the initial visit and their evolution over the follow-up period. Values are expressed as means ± SEM; BMIc = body mass index category.

**Figure 2 nutrients-14-05191-f002:**
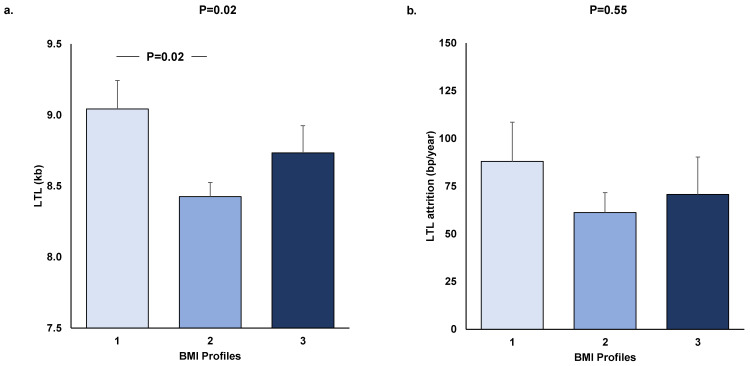
Baseline LTL (**a**) and LTL attrition (**b**) in the three BMI profiles.LTL values were age- and sex-adjusted, and LTL attrition values were age-, sex-, and initial LTL-adjusted. Values are expressed as means ± SEM, and p is derived from ANOVA. Tukey–Kramer post hoc test was used for two-by-two comparisons. LTL = leukocyte telomere length; kb = kilobase; BMI = body mass index; bp = base pair.

**Figure 3 nutrients-14-05191-f003:**
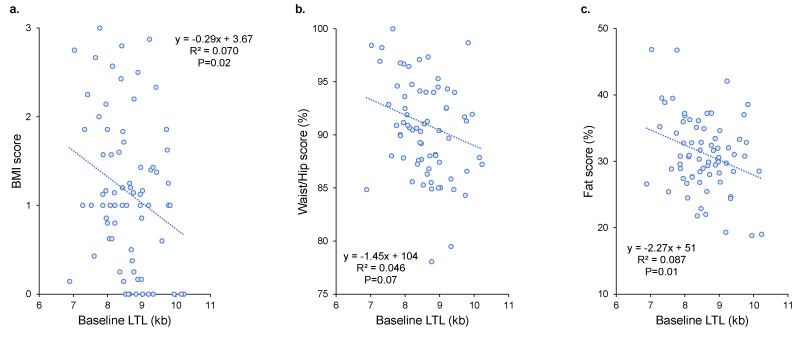
Relationships between baseline LTL and BMI (**a**), waist/hip (**b**), and fat percentage (**c**) scores. R^2^ and p are from Pearson’s correlation. LTL = leukocyte telomere length; kb = kilobase; BMI = body mass index.

**Figure 4 nutrients-14-05191-f004:**
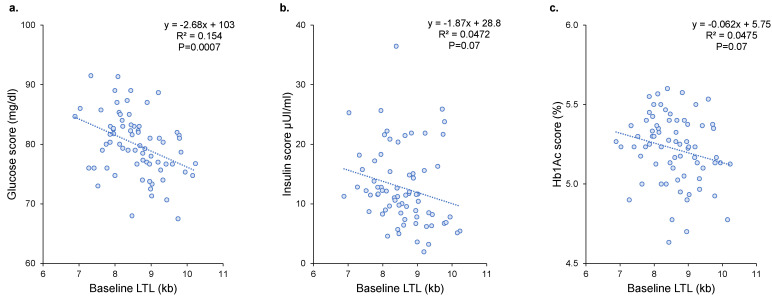
Relationships between baseline LTL and glucose (**a**), insulin (**b**), and Hb1Ac (**c**) scores. R^2^ and p are from Pearson’s correlation. LTL = leukocyte telomere length; kb = kilobase.

**Table 1 nutrients-14-05191-t001:** Characteristics of the participants according to BMI profiles.

	All	Profile 1	Profile 2	Profile 3	One Way ANOVA *p*	Trend *p*
Number	73	12	48	13		
Females (%)	66%	58%	75%	38% #	0.04	0.26
Age at baseline (years)	7.6 ± 2.0	7.1 ± 2.2	7.7 ± 1.9	7.8 ± 2.0	0.64	0.38
Age at follow-up (years)	10.3 ± 2.1	9.9 ± 2.3	10.4 ± 2.1	10.5 ± 1.9	0.74	0.47
Follow-up (years)	2.8 ± 0.7	2.8 ± 0.7	2.8 ± 0.8	2.7 ± 0.5	0.91	0.75
Baseline LTL (kb)	8.58 ± 0.75	9.06 ± 0.74	8.46 ± 0.72 *	8.61 ± 0.71	0.04	0.13
Follow-up LTL (kb)	8.40 ± 0.78	8.85 ± 0.72	8.29 ± 0.78	8.41 ± 0.78	0.08	0.16
LTL attrition (bp/year)	67 ± 69	83 ± 56	63 ± 70	69 ± 80	0.67	0.60
Baseline BMIc	1.70 ± 1.13	0.00	2.13 ± 0.89	1.69 ± 0.95	-	-
Baseline glucose (mg/dl)	78 ± 9	76 ± 7	78 ± 9	78 ± 13	0.78	0.63
Baseline insulin (µUI/mL)	10.1 ± 5.6	6.5 ± 2.8	11.1 ± 5.6 *	10.2 ± 6.5	0.049	0.09
Baseline HbA1c (%)	5.22 ± 0.24	5.09 ± 0.34	5.24 ± 0.23	5.28 ± 0.16	0.14	0.06
Baseline waist (cm)	72 ± 11	58 ± 8	74 ± 9 ***	79 ± 9 ***	<0.0001	<0.0001
Baseline hip (cm)	78 ± 12	65 ± 10	79 ± 11 ***	83 ± 10 ***	<0.0001	<0.0001
Baseline waist/hip (%)	93 ± 8	90 ± 5	93 ± 9	94 ± 7	0.24	0.17
Baseline fat percentage (%)	31.5 ± 6.6	24.3 ± 4.6	33.3 ± 5.2 ***	32.9 ± 8.2 **	0.0003	0.002
Mother diabetes mellitus during pregnancy	23%	10%	27%	20%	0.51	0.60

Values are expressed as means ± SD or %; LTL = leukocyte telomere length; kb = kilobase; bp = base pair; BMIc = body mass index category; Hb1Ac = hemoglobin A1c.; Tukey–Kramer post hoc test was used for comparison between two profiles. # *p* < 0.05 vs. Profile 2, * *p* < 0.05 vs. Profile 1; ** *p* < 0.01 vs. Profile 1, and *** *p* < 0.005 vs. Profile 1. In the absence of a normal distribution, Kruskal–Wallis one-way ANOVA was used for baseline insulin and baseline waist/hip

**Table 2 nutrients-14-05191-t002:** Multivariate models analyzing the effects of LTL and covariates on obesity scores.

Dependent Variable	Independent Variable	Regression Coefficient ± Standard Error	R^2^	*p*
BMI score	LTL baseline (1 kb)	−0.29 ± 0.13	7.0%	0.02
	Age (1 year)	-	-	0.18
	Female (yes)	-	-	0.26
	Model		7.0%	
Waist/hip score (%)	LTL baseline (1 kb)	−2.22 ± 0.77	9.8%	0.005
	Age (1 year)	−0.99 ± 0.29	13.8%	0.001
	Female (yes)	-	-	0.13
	Model		18.3%	
Fat score (%)	LTL baseline (1 kb)	−2.60 ± 0.87	11.1%	0.004
	Age (1 year)	-	-	0.62
	Female (yes)	3.01 ± 1.34	6.2%	0.03
	Model		14.9%	

LTL = leukocyte telomere length; kb = kilobase; BMI = body mass index.

**Table 3 nutrients-14-05191-t003:** Multivariate models analyzing the effects of LTL and covariates on diabetes scores.

Dependent Variable	Independent Variable	Regression Coefficient ± Standard Error	R^2^	*p*
Glucose score	LTL baseline (1 kb)	−2.68 ± 0.76	15.4%	0.0007
	Age (1 year)	-	-	0.62
	Female (yes)	-	-	0.71
	Model		15.4%	
Insulin score	LTL baseline (1 kb)	-	-	0.11
	Age (1 year)	1.17 ± 0.36	13.0%	0.002
	Female (yes)	3.16 ± 1.48	5.4%	0.04
	Model		17.9%	
Hb1Ac score	LTL baseline (1 kb)	−0.061 ± 0.033	4.7%	0.07
	Age (1 year)	-	-	0.81
	Female (yes)	-	-	0.59
	Model		4.7%	

LTL = leukocyte telomere length; kb = kilobase.

## Data Availability

The data presented in this study are available from the corresponding author upon reasonable request.

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
