# Peer review of "Longitudinal Association of Telomere Dynamics with Obesity and Metabolic Disorders in Young Children"

_nutrients, 2022, doi:10.3390/nu14235191_

Round 1

Reviewer 1 Report

In the well written manuscript entitled "Association of telomere dynamics with obesity and metabolic disorders in young children", Toupance et al. studied blood leukocyte telomere length (attrition) in a population of children with obesity as well as controls. Though the population size is limited, this is partially compensated by the very good technical error on the telomere length measurements (<2%). The authors report a significant association between baseline telomere length (TL) and several obesity as well as metabolic disorder associated variables. The respective associations with TL attrition were however not significant.

I have no substantial remarks regarding the baseline telomere length results, which are clearly consistent, and largely confirmed by the measurements and associated cross-sectional analyses at follow up. Though I agree that these results are very relevant, I recommended to add some clearly pertinent literature (particularly by the group of Tim Nawrot, see e.g. Clemente et al. Sci Rep 2019) to the introduction and/or discussion section. Also, it may be added that the results in fact support the notion that variable telomere attrition rates as well as lifestyle between individuals may further obscure the TL - obesity association in later life. This is particularly clear in adults (required large-scale meta-analysis to obtain final proof, cf. Gielen et al. Am J Clin Nut 2018), but is also supported by the slightly weaker follow up associations in the current study.

Regarding the telomere length attrition results however, I do not readily consider this to be convincing proof yet. As indicated by the authors, the limited sample size will particularly lead to lower power for statistical analyses of attrition estimates, given the latter's inherently higher measurement error. I therefore suggest to also provide beta's for attrition rate models (currently, only p-values are mentioned), which can be more easily compared to the cross-sectional results than basic p-values. However it should also be noted that telomere length attrition is slower in adults compared to children. Therefore, a potential major issue is the fact that telomere length attrition may be substantially faster in the youngest children compared to the oldest ones in the study at hand, obscuring the more limited (hypothetical) impact of obesity on attrition. This issue can be easily evaluated using the data from the study, i.e. by modelling TL attrition as a function of average age (not baseline age, as this may induce "regression to the mean" effects), and I recommend to do so. Nevertheless, even if this phenomenon would be present, detecting and adjusting for it may be hard given the limited sample size as well as potentially non-linear effects. Independent of the outcome, I therefore particularly suggest to adequately discuss this in the manuscript. 

In conclusion, I largely agree with the authors' conclusions and concur that they provide relevant novel insight. For the telomere attrition results however, I suggest some additional analyses and discussion.

Author Response

Please see the attachment. Our answers are in blue.

Reviewer 2 Report

I found the study to be informative, well-written, and well-researched in general. In light of the alarming increase in metabolic disorders among children globally, and the need to modify lifestyles and diets, these studies are necessary.

In the manuscript “Association of telomere dynamics with obesity and metabolic 2 disorders in young children” the authors investigated the shortening of telomere length and attritions in relation to metabolic disorders such as obesity and diabetes mellitus type 2.

I have a few minor comments for the authors to improve the manuscript

1.     Although standard abbreviates are used, it is recommended that the expanded forms be used when using them for the first time.

a.      Methods section 2.2 Page 2 – Line 69 WHO

b.     Methods section 2.3 Page 2 – Line 84 USDA

2.     2.1 (Patients) mentions the study period of four years, but it would be useful to provide information regarding which year to year the study was conducted. It is likely that the study was conducted during the Covid 19 lockdown period, given the length of the study.

3.     Page 2 Line 57, the exclusion criteria were written, preferably inclusion criteria can also be pointed out which gives more clarity.

4.     Page 2, Line 69, for the “WHO STEPS” the added reference could be informative.

5.     Page 2, Line 78, site of blood collection could be added.

6.     Page 2 Line 94, the author has pointed out that hematological investigations were determined, but nowhere in the manuscript, they didn’t discuss.

7.     Page 5, Table 1 the expansion of HbA1c might be included in the legend.

8.     Introduction is less informative and in the Discussion section, the parameters studied can be discussed efficiently.

9.     Number of adipokines, including TNF-α, IL-6,8, and 10, leptin, and adiponectin affects insulin sensitivity through the modulation of insulin signaling. Why did not use a biomarker to study insulin sensitivity/obesity/ DM?

Author Response

(The authors gave the same response as above.)

Reviewer 3 Report

This is a prospective study including 73 participants. Various characteristics were recorded and compared among risk of BMI. Stratifications by the eating styles results were made regarding different BMI characteristic. They found that short telomeres contribute to the development of obesity and metabolic disorders very early in life, which can have a major impact on health..

This is an interesting study with some new findings in this area of research. The sample size of subjects is small for analysis. However, I nevertheless have the following comments that required to be addressed.

1.     The study design should be specified in title of this study. The authors should clarify this concern.

2.     The statistical methods used and described very well. Why use ANOVA without normality test?

3.     How does the authors to determine the sample size of this study? Please use power analysis to statement adequate sample size in this study.

4.     It seems the R2 was not satisfied for researchers. The authors should highlight novelty in this study. What do we newly learned from this study?

5.     Lastly, the authors only briefly discuss limitations, but didn’t acknowledging that the one of limitation include the prospective study. he authors should clarify this concern.

Author Response

(The authors gave the same response as above.)
